# Residues of Selected Anticonvulsive Drugs in Surface Waters of the Elbe River Basin (Czech Republic)

**Martin Ferencik** [1,2], **Jana Blahova** [3,*], **Jana Schovankova** [1], **Zuzana Siroka** [3], **Zdenka Svobodova** [3], **Vit Kodes** [4], **Karla Stepankova** [3] and **Pavla Lakdawala** [3]

1   Elbe River Basin, State Enterprise, 500 03 Hradec Kralove, Czech Republic
2   Institute of Environmental and Chemical Engineering, Faculty of Chemical Technology, University of Pardubice, 532 10 Pardubice, Czech Republic
3   Department of Animal Protection and Welfare & Veterinary Public Health, Faculty of Veterinary Hygiene and Ecology, University of Veterinary Sciences Brno, 612 42 Brno, Czech Republic
4   Czech Hydrometeorological Institute, 143 00 Prague, Czech Republic
*   Correspondence: blahovaj@vfu.cz

**Abstract:** Anticonvulsants are pharmaceuticals used for epilepsy treatment, pain syndromes therapy and for various psychiatric indications. They enter the aquatic environment mainly through wastewater and were found to cause both biochemical and behavioral changes in aquatic biota. Because the consumption of anticonvulsive drugs is quite high, their monitoring in the aquatic environment is needed. The Elbe River basin is the fourth largest in Europe; the Elbe flows into the North Sea and therefore its contamination is of international importance. The aim of the present study was to determine the presence and concentrations of anticonvulsant pharmaceuticals (carbamazepine, lamotrigine and gabapentin) and their analogues (gabapentin-lactam) in water samples obtained from the Elbe River and its tributaries located in the Czech Republic, as well as to evaluate their correlations with flow rates. The results of this study show that the selected drugs are present in the surface water of the Elbe River in tens to hundreds of ng/L, with the highest measured concentrations for gabapentin. Our results also indicate that the further the sampling point from the Elbe spring is, the higher the concentrations of monitored pharmaceuticals are. Moreover, small tributaries are significantly more contaminated due to their low flow rates with the exceptions of streams flowing from preserved natural sites. The results of the monitoring highlight the importance of building wastewater treatment plants at the municipalities where they are still not present with focus on technology that would be able to decompose substances with negative removal efficiency.

**Keywords:** gabapentin; gabapentin-lactam; carbamazepine; lamotrigine; aquatic pollution; anticonvulsants

## 1. Introduction

Anticonvulsants are human and veterinary pharmaceuticals used for many purposes. Besides their main domain, which is epilepsy treatment (antiepileptic drugs), anticonvulsants are also widely used for pain syndromes therapy and in several psychiatric indications (e.g., panic and anxiety disorders, dementia, schizophrenia), as well as certain personality disorders (e.g., bipolar disorder) [1]. Antiepileptic drugs act on diverse molecular targets to selectively modify the excitability of neurons and prevent seizure-related firing, preferentially without influencing non-epileptic neuronal activity. This selective modification of function can also be beneficial in the management of other diseases connected with similar signaling pathways in the brain. Ion channels are important for signaling; the rates of channel opening and closing are influenced by the voltage changes and binding of different neurotransmitters [2]. These processes may by modulated both in patients and in non-target organisms after the drugs are excreted and enter the environment. Anticonvulsive substances are known to cause both biochemical and behavioral changes, e.g., in fish [3,4] and to accumulate in their tissues and enter the food chain [5,6].

Thus, monitoring of these pharmaceuticals in different parts of ecosystems and biota are necessary. There are several studies revealing these drugs in the aquatic environment (water and sediment), soil and plants. The main source of entry to these systems is wastewater, mainly from large urban areas [7–12] and incomplete removal in standard wastewater treatment plants. Many of these substances are quite resistant to several bio(degradation) procedures, some of them even to advanced processes [13–17]. Pharmaceutical residues are not only a result of continuous release, but due to their persistence and resistance they also belong to so-called core micropollutants with permanent occurrence [11] and ubiquitous spreading (gabapentin was found also in Antarctic phytoplankton) [18].

The Elbe River and its tributaries are a major water resource in the Czech Republic and downstream also in Germany. The Elbe flows into the North Sea approximately 100 km northwest of Hamburg. The Elbe River basin is the fourth largest in Europe, draining 148,270 km$^2$ [19,20].

The aim of this study was to determine the presence and concentrations of anticonvulsant pharmaceuticals (carbamazepine, lamotrigine, and gabapentin) and the analogue gabapentin (i.e., gabapentin-lactam) in water samples obtained from 10 sampling locations in the Elbe River and its tributaries located in the Czech Republic, and to evaluate their correlations with the flow rates at these sampling sites in the year 2021 in order to see whether tributaries with small flow rates possess higher ecotoxicity risks (due to higher concentrations) for aquatic organisms.

## 2. Materials and Methods

### 2.1. Chemicals and Reagents

Four anticonvulsive pharmaceuticals and their analogues were monitored in the Elbe River basin at 10 sampling sites. Standards of them were purchased as follows: carbamazepine (CAS 298-46-4, $C_{15}H_{12}N_2O$) was obtained from Dr. Ehrenstorfer (Augsburg, Germany); gabapentin (CAS 60142-96-3, $C_9H_{17}NO_2$) from Toronto Research Chemicals (North York, ON, Canada); gabapentin-lactam (gabapentin EP Impurity A, CAS 64744-50-9, $C_9H_{15}NO$) and lamotrigine (CAS 84057-84-1, $C_9H_7Cl_2N_5$) were provided by European Pharmacopoeia Reference Standards (Strasbourg, France). Two isotopic labelled analogues: carbamazepine-[2H15N] and gabapentin-[2H4] were provided by Toronto Research Chemicals (North York, ON, Canada). Solvents used were methanol CHROMASOLV™ LC-MS from Honeywell, Riedel-de Haën™ (Germany), and ammonium acetate for mass spectrometry LiChropur™ (eluent additive for LC-MS) from Merck Millipore (Darmstadt, Germany). Water used for mobile phase preparation and preparation of standard calibration solutions was produced by ultrapure water purification system SG Ultra Clear TW UV Plus™ (Hamburg, Germany).

### 2.2. Sample Collection and Sampling Sites

During 2021, the grab samples were taken every month from 5 localities of the Elbe River and from 5 localities placed in the tributaries of the Elbe River (Figures 1 and 2). The sites selected for monitoring in this study are part of the monitoring profiles of the national surface water situational monitoring network, which have been determined in accordance with Directive 2000/60/EC of the European Parliament and of the Council establishing a framework for Community action in the field of water policy. Sampling points were selected as follows: Elbe Debrné—foothill area with little anthropogenic load; sampling point Elbe Hradec Králové—located above the confluence with the Orlice River and above the outlet of the Hradec Králové wastewater treatment plant; sampling point Elbe Valy—below the industrial agglomeration of Pardubice (chemical industry) and below the outlet of the Pardubice wastewater treatment plant; sampling point Elbe Obříství—above the confluence with the Vltava River; sampling point Elbe Děčín above the outlet to Germany; sampling points of important tributaries (Cidlina, Mrlina, Doubrava)—mainly for agricultural use with smaller human settlements, often without wastewater treatment plants; Orlice Nepasice and Jizera Tuřice streams with a mixed character (agricultural, industrial with outlet to the wastewater treatment plants of district towns) and having a

significant source of clean water from mountainous areas—both used as sources of raw water for the production of drinking water (Orlice for the city of Hradec Králové and its surroundings, Jizera for the Prague agglomeration).

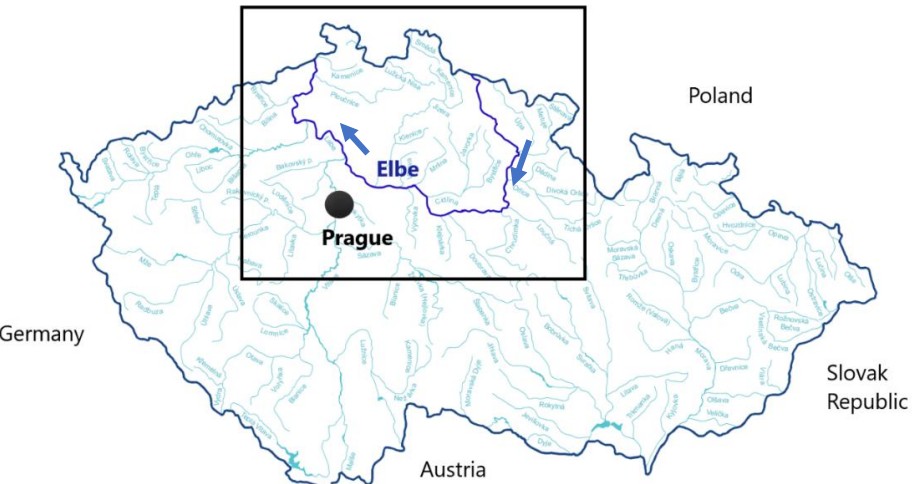

**Figure 1.** Map of the Elbe River basin in the Czech Republic. Blue arrows show the direction of the River Elbe flow.

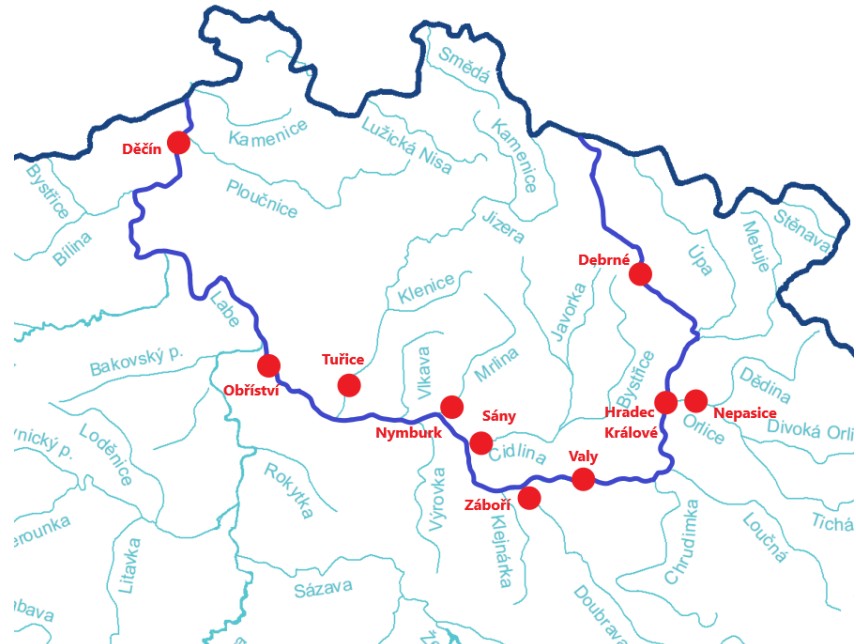

**Figure 2.** Map of sampling sites of the Elbe River basin.

Water samples were taken in 1-L glass bottles, cooled and transported to the laboratory for analysis. The water was taken in the water column in such a way that it was not affected by the upper layer (i.e., floating objects and any undissolved organic substances with a lower density than water) and the river sediments were not disturbed. Average daily discharge rates at individual localities in the days of sampling were used to calculate flux values. The flows used for flux calculations were either directly measured at Czech Hydrometeorological Institute water gauging stations (in cases where the samples were taken at the water gauging stations) or calculated from discharge data of nearby Czech Hydrometeorological Institute water gauging stations using the Czech Hydrometeorological Institute internal method for discharge calculations at ungauged sites (in cases where the

samples were taken at either downstream or upstream water gauging stations) based on the drainage area ratio method [21,22].

### 2.3. Sample and Calibration Standards Preparation

The calibration standard solutions of the pharmaceutical mixtures (0, 5, 10, 20, 50, 100, 250, 1000 and 2000 ng/L) were prepared. Each concentration level was measured in triplicate and linear calibration curves were constructed. Correlation coefficients were higher than 0.995. Carbamazepine and gabapentin were corrected by their corresponding isotopic analogue internal standards and the others were corrected by the standard addition method. The quantification limits (LOQ) were as follows: carbamazepine 5 ng/L, gabapentin 20 ng/L, gabapentin-lactam 10 ng/L, lamotrigine 50 ng/L. Samples were filtered through a syringe filter 17 mm in diameter, 0.2-µL porosity with regenerated cellulose membrane (Thermo Fisher Scientific, Waltham, MA, USA) into 15-mL clear glass vials with a polytetrafluoroethylene (PTFE) seal screw-top (Supelco, Bellefonte, PA, USA), which were subsequently stored at 4 °C. After 24 h, 1 mL of each filtered sample was dispensed by a 1-mL gas-tight Hamilton syringe in two 2-mL TruView™ vials (Thermo Fisher Scientific, Waltham, MA, USA) with a slit PTFE/silicone septum for measurement in the autosampler. Then, 50 µL of the internal standard stock solution were added to each vial with 1 mL of the sample. One of them was assigned a plus sign due to the addition of 10 µL of the working standard solution (10 µL of the solution at a concentration of 10 µg/L, where the final concentration of a spiked sample was 100 ng/L). Both vials were marked with the specific sample identifiers and were subsequently entered into the measurement spreadsheet of MassLynx computer software (Waters Corporation, Milford, MA, USA) with this code. Immediately after preparation, all the vials were capped and prepared for analysis.

### 2.4. Instrumentation

An Ultra Performance Liquid Chromatograph (UPLC™) Acquity™ Waters (Milford, MA, USA) with a Triple Quadrupole Mass Spectrometer Waters Premier XE Micromass™ MS Technologies (Manchester, UK) operated with software MassLynx™ V4.1 were used for the analyses. The electrospray ion source (ESI) operated in the positive ion mode at a capillary voltage of 0.7 kV, an ion source temperature of 120 °C, a desolvation temperature of 350 °C and a flow of nitrogen supplied by the nitrogen generator Infinity 5020 (Peak Scientific Instruments, Inchinnan, Scotland, UK), a desolvation gas flow at 700 L/hr and cone gas flow at 100 L/hr. The collision gas was argon (Ar purity Premier, Air Products, Děčín, Czech Republic) used at a pressure of approximately 0.2 Pa.

The chromatographic separation was achieved on an Acquity UPLC™ HSS T3 Column, 1.8 µm, 2.1 mm × 100 mm (Waters, Milford, MA, USA) and an Acquity UPLC™ HSS T3 VanGuard™ Pre-column, 1.8 µm, 2.1 mm × 5 mm (Waters, Milford, MA, USA) using gradient elution with a flow rate of 0.25 mL/min at 40 °C. The mobile phase A was methanol with 5 mmol/L ammonium acetate, whereas the mobile phase B consisted of 5% vol. methanol with 5 mmol/L of ammonium acetate. The linear gradient was as follows: isocratic 0–3 min 99.9% B, linear 3–16 min 99.9–0.1% B, isocratic hold 16–18.3 min 0.1% B and linear gradient 18.3–19.5 min 0.1–99.9% B. Total analysis time from injection to injection was 24.5 min. Injection volume was 250 µL.

The triple quadrupole mass spectrometer was operated in multiple reaction mode (MRM), where the dwell time of each transition was 10 ms and two transitions (precursor > product) with optimized values of cone voltage (CV) and collision energy (CE). Transitions with corresponding (CV, CE) for measured substances were the following: carbamazepine 237.2 > 194.1 (32, 21) and 237.2 > 179 (32, 34); carbamazepine-[2H15N] 241.9 > 197.9 (28, 21) and 241.9 > 181.5 (28, 37); gabapentin 172.0 > 154.0 (23, 13) and 172.0 > 136.9 (23, 12); gabapentin-[2H4] 176.1 > 158.0 (20, 14) and 176.1 > 96.9 (20, 24); gabapentin-lactam 154.1 > 94.9 (40, 20) and 154.1 > 67.0 (40, 26); lamotrigine 256.2 > 211 (45, 25) and 256.2 > 186.7 (46, 23). Chromatograms from the analysis are provided in Figure S1.

### 2.5. Statistical Analysis

Statistical analysis was performed using the statistical software Unistat for Excel 6.5 (London, England). For the evaluation of differences in individual concentrations of the selected anticonvulsants among localities, data were tested using the Shapiro–Wilk test. Levene's test was used to evaluate the normality and homogeneity of variances across localities. As normality of the data was not achieved, data were further analyzed using the nonparametric multi-sample median test. In cases where an individual anticonvulsive drug was not detected in a location, half of the detection limit of this substance was used for statistical analysis (LODs determined from a 3:1 signal-to-noise ratio are as follows: carbamazepine LOD = 1 ng/L, gabapentin LOD = 5 ng/L, gabapentin lactam LOD = 2 ng/L, lamotrigine LOD = 10 ng/L). In addition, Spearman's rank-order correlation for concentrations of individual anticonvulsive drugs and flow rates at monitored localities, as well as for years and delivery of drugs to pharmacies/medical facilities was performed. Significance was accepted at $p < 0.05$.

## 3. Results

Based on the consumption data of anticonvulsants in the Czech Republic (Table 1), it is obvious that it is quite high, with gabapentin being the most frequently used anticonvulsive drug among those selected for this study.

**Table 1.** Trend of usage (based on the delivery of drugs to pharmacies and medical facilities) of selected anticonvulsive pharmaceuticals in the Czech Republic during years 2011–2020. Data were obtained from the State Institute for Drug Control, Czech Republic (www.sukl.cz). Relationship between years and usage was analyzed using the Spearman's rank-order correlation ($r_s$).

| | Delivery [kg/year] | | |
|---|---|---|---|
| Year | Gabapentin | Carbamazepine | Lamotrigine |
| 2011 | 8420 | 5465 | 1366 |
| 2012 | 9333 | 5171 | 1426 |
| 2013 | 10,429 | 5959 | 1472 |
| 2014 | 10,429 | 4807 | 1472 |
| 2015 | 13,141 | 4643 | 1589 |
| 2016 | 13,746 | 4328 | 1678 |
| 2017 | 14,360 | 4215 | 1705 |
| 2018 | 14,558 | 3929 | 1779 |
| 2019 | 16,127 | 3736 | 1806 |
| 2020 | 15,390 | 3540 | 1866 |
| $r_s$ | 0.9848 | −0.9636 | 0.997 |
| *p* value | 0.000 | 0.000 | 0.000 |

Statistically, highly significant positive correlations ($p = 0.000$) over time were confirmed for gabapentin and lamotrigine, indicating their increasing consumption compared to carbamazepine, where the trend is reversed. The data on anticonvulsive drug residue concentrations at the monitored localities throughout the year 2021 are given in Figures 3–6 with gabapentin expectedly (based on the data given in Table 1) reaching the highest residue content out of the 4 investigated substances. However, carbamazepine, gabapentin-lactam and lamotrigine concentrations were measured in the tens of ng/L, and in the case of gabapentin it was hundreds of ng/L. As evident from Figures 3–6, anticonvulsant residue content substantially differs among the months. Flow rates at monitored sites throughout the year are also provided in Figure 7. To see whether there is a connection between residue concentrations and flow rates, a correlation was determined with the results of this analysis given in Table 2. In most cases (except for gabapentin), the value given in Table 2 was found to be negative, which indicates the higher the flow rate the lower the concentration (because of higher dilution). Median concentrations of drugs selected for this study are given in Figures 8–11. In these figures, significant differences between the

localities are indicated. The results of this study show that the further the sampling point from the Elbe River spring is, the higher the concentrations of monitored pharmaceuticals are. This makes the sampling site Děčín the most contaminated site, directly on the Elbe River. Also, several small tributaries, such as Doubrava, Cidlina and Mrlina, were found to be significantly contaminated due to their low flow rates. In relation to the increased flow rates in streams for the first half of the year 2021 (Figure 7), lamotrigine concentrations were found to be below the level of detection during this period in almost all the sampling sites (Figure 6). The data on mass flux (Figure 12) support the above-described data presented in Figures 3–11. A list of the most important wastewater treatment plants located upstream from the indicated sampling points is provided in the Supplementary Material (Table S1).

**Table 2.** Results of the Spearman's rank-order correlation for concentrations of individual anticonvulsive drugs and flow rate at monitored localities during the year 2021. The statistically significant Spearman´s rank coefficient ($p < 0.05$) is indicated in boldface.

| Localities | Gabapentin | Gabapentin-Lactam | Carbamazepine | Lamotrigine |
|---|---|---|---|---|
| Elbe (Debrné) | −0.455 | **−0.6862** | **−0.946** | **−0.587** |
| Elbe (Hradec Králové) | −0.288 | **−0.9062** | **−0.885** | **−0.707** |
| Elbe (Valy) | 0.063 | **−0.7461** | **−0.787** | **−0.778** |
| Elbe (Obříství) | 0.049 | −0.4471 | **−0.658** | **−0.573** |
| Elbe (Děčín) | −0.029 | −0.2853 | **−0.685** | **−0.593** |
| Orlice (Nepasice) | −0.203 | **−0.6877** | **−0.968** | **−0.670** |
| Doubrava (Záboří) | −0.028 | **−0.8511** | **−0.844** | **−0.843** |
| Cidlina (Sány) | −0.140 | **−0.6154** | **−0.818** | **−0.705** |
| Mrlina (Nymburk) | **−0.846** | **−0.9161** | **−0.865** | **−0.848** |
| Jizera (Tuřice) | 0.132 | −0.4472 | **−0.656** | −0.330 |

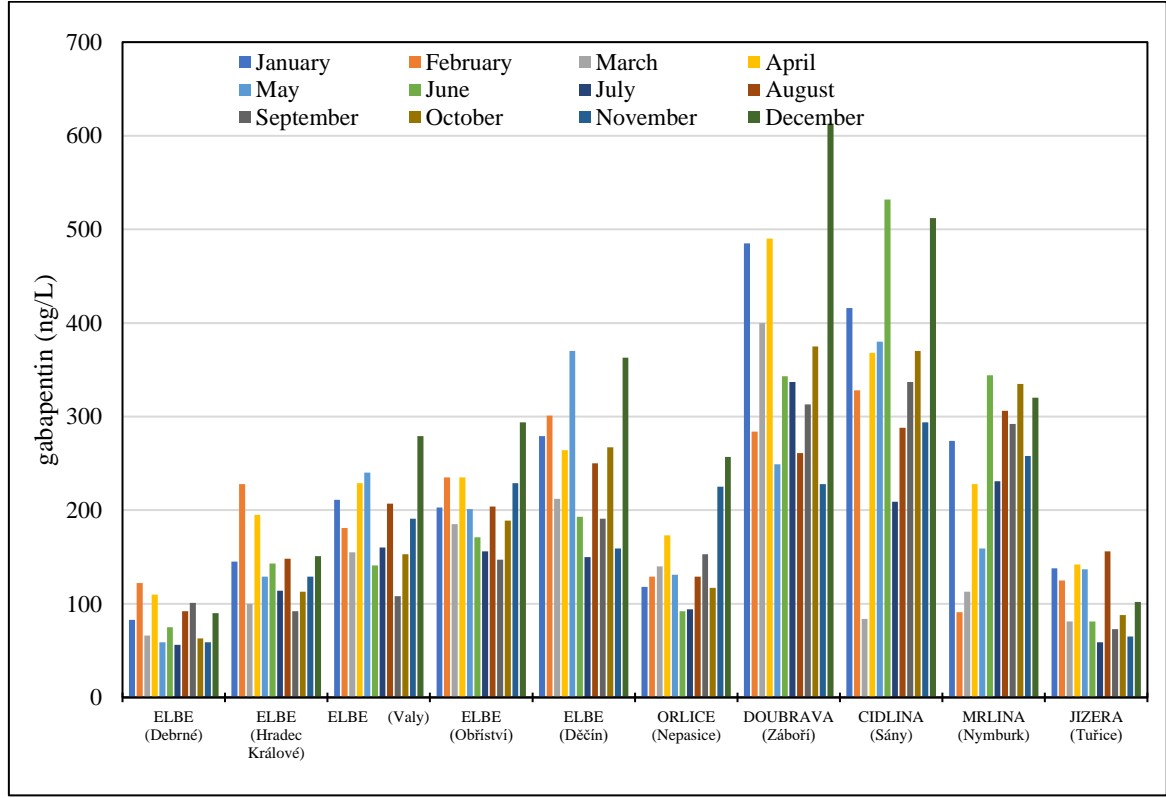

**Figure 3.** Residues of gabapentin in surface water of the Elbe River basin.

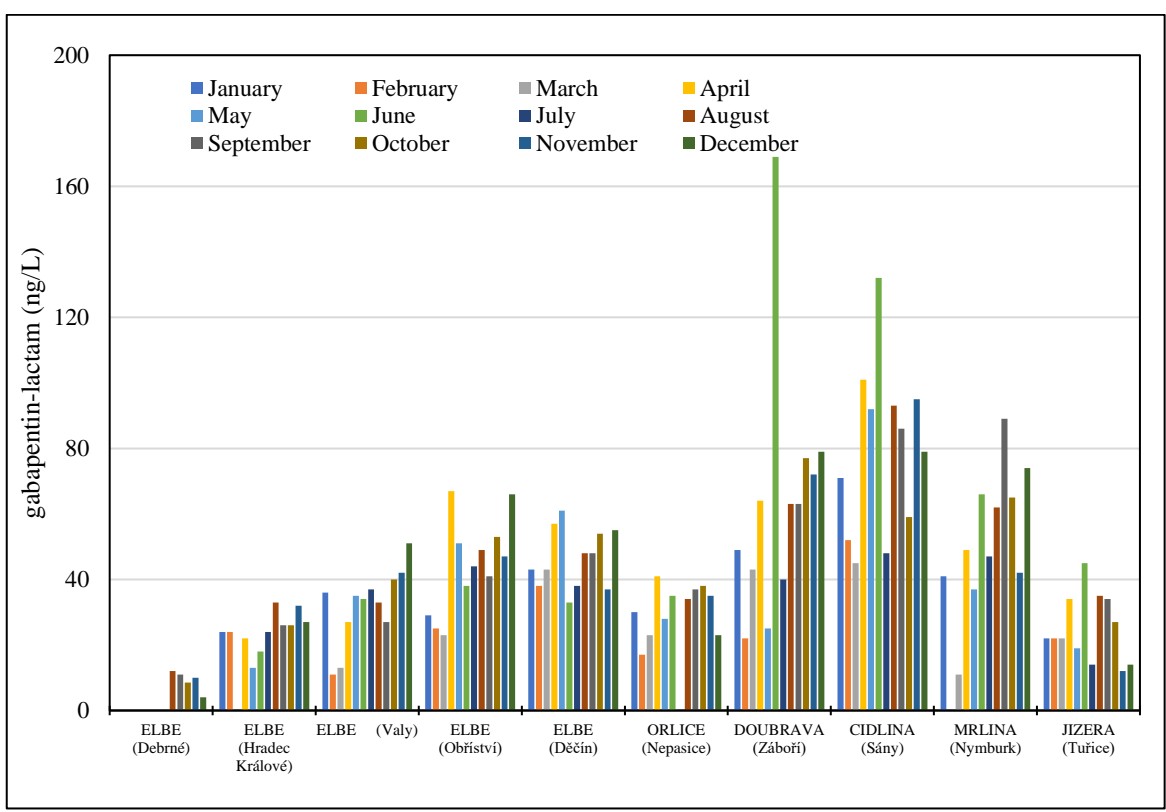

**Figure 4.** Residues of gabapentin-lactam in surface water of the Elbe River basin.

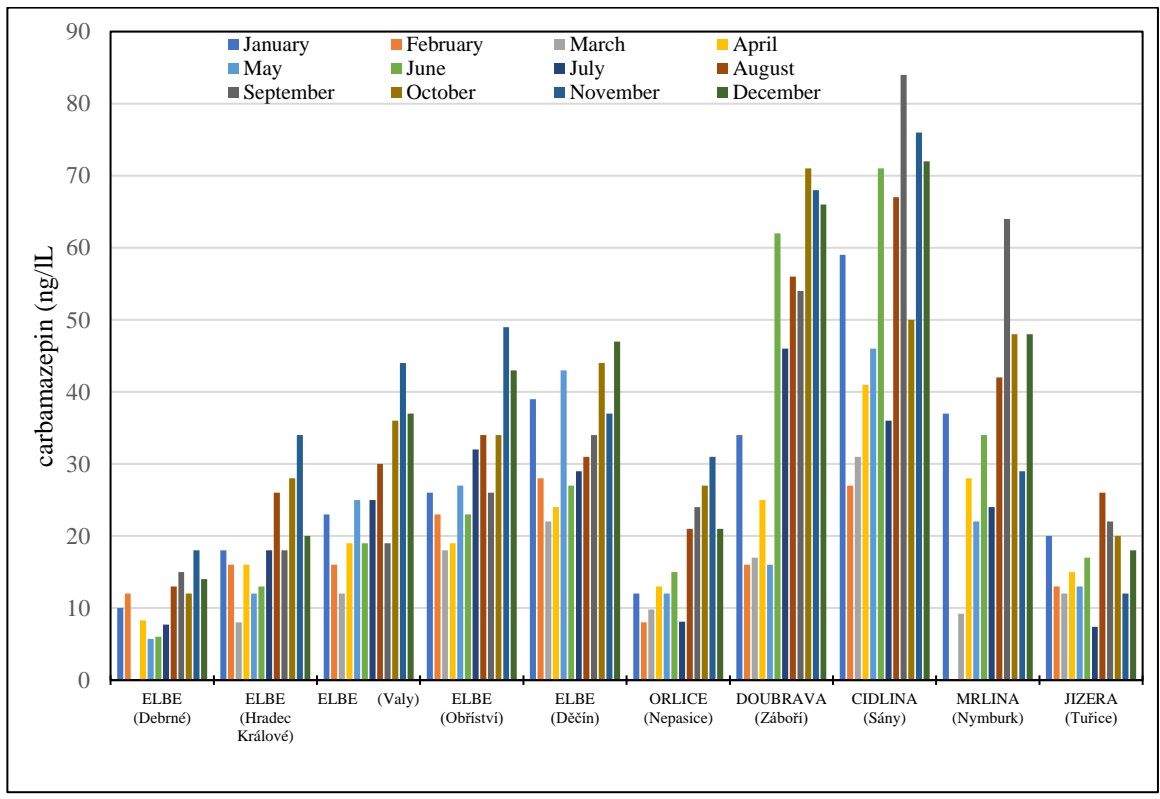

**Figure 5.** Residues of carbamazepine in surface water of the Elbe River basin.

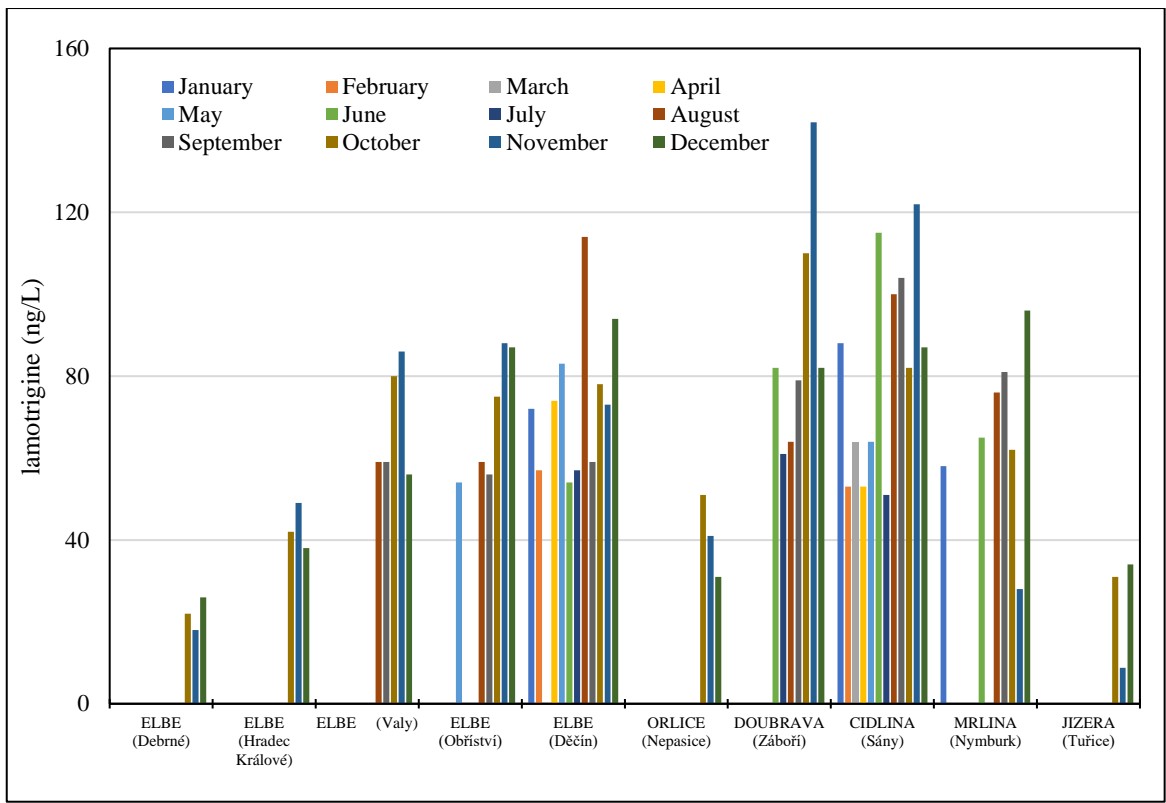

**Figure 6.** Residues of lamotrigine in surface water of the Elbe River basin.

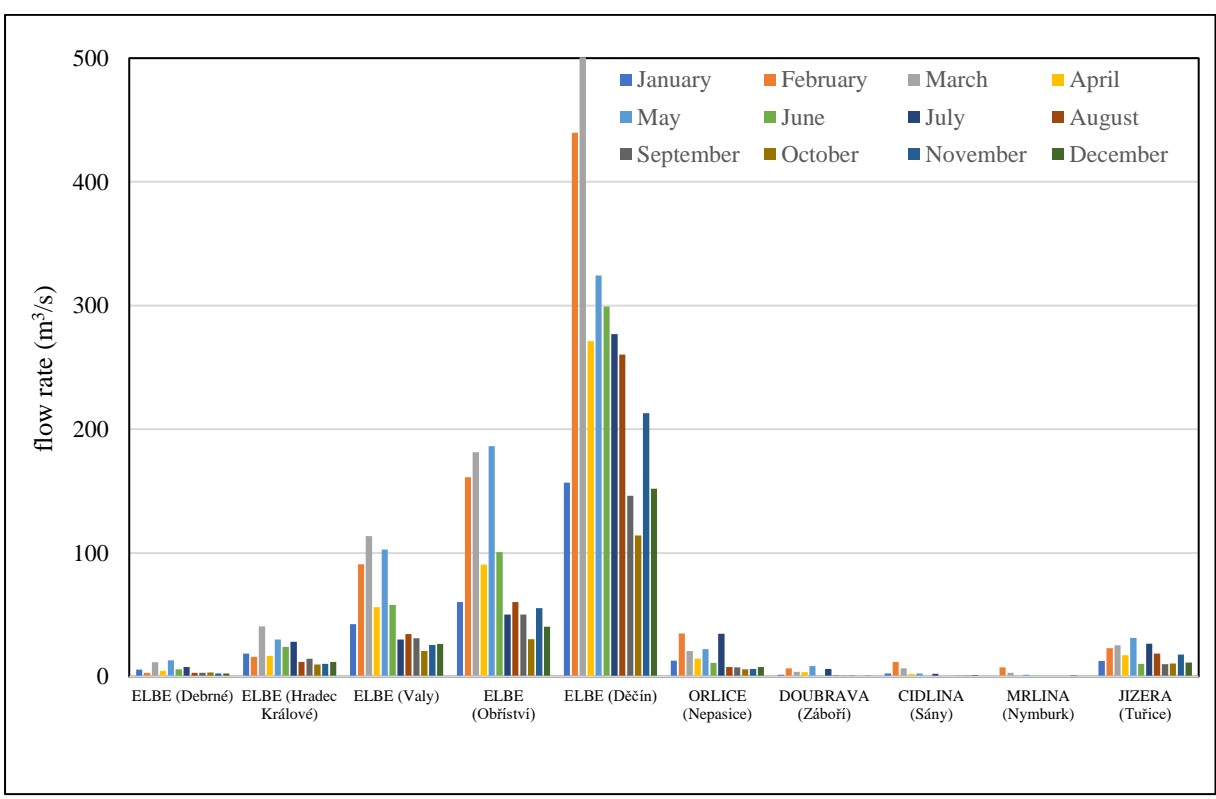

**Figure 7.** Daily average flow rate at monitored sites of the Elbe River basin.

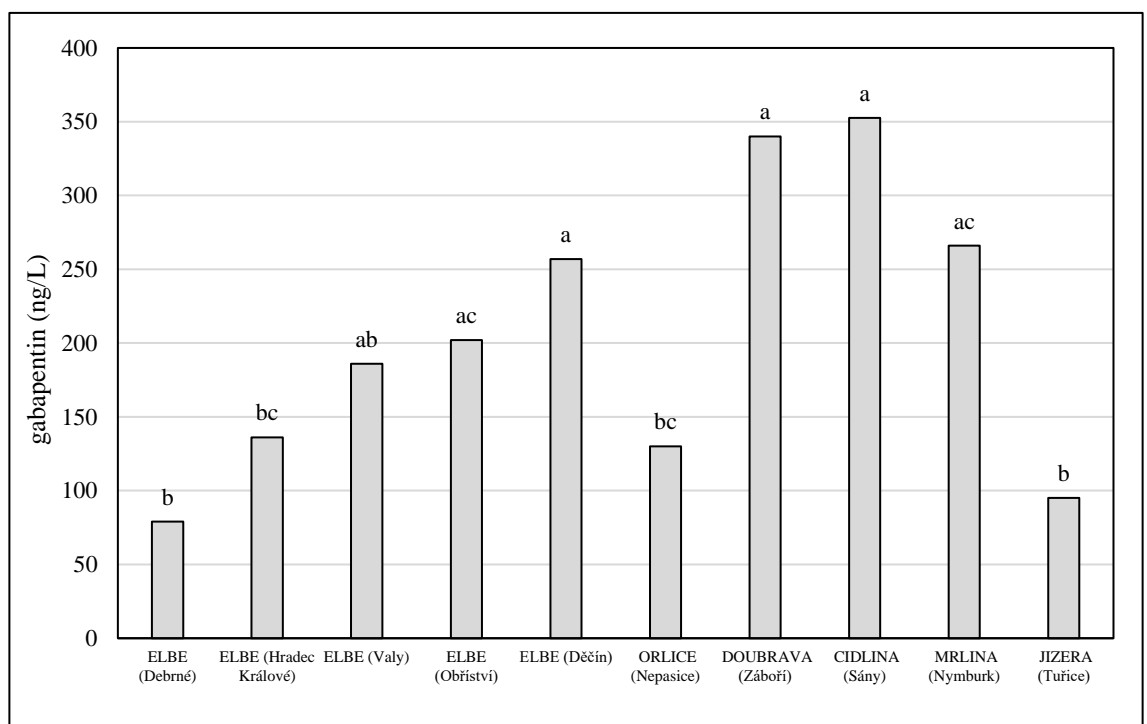

**Figure 8.** Median values of gabapentin concentration at the sampled locations during the year 2021. Significant differences among localities are indicated by different alphabetical superscripts ($p < 0.05$).

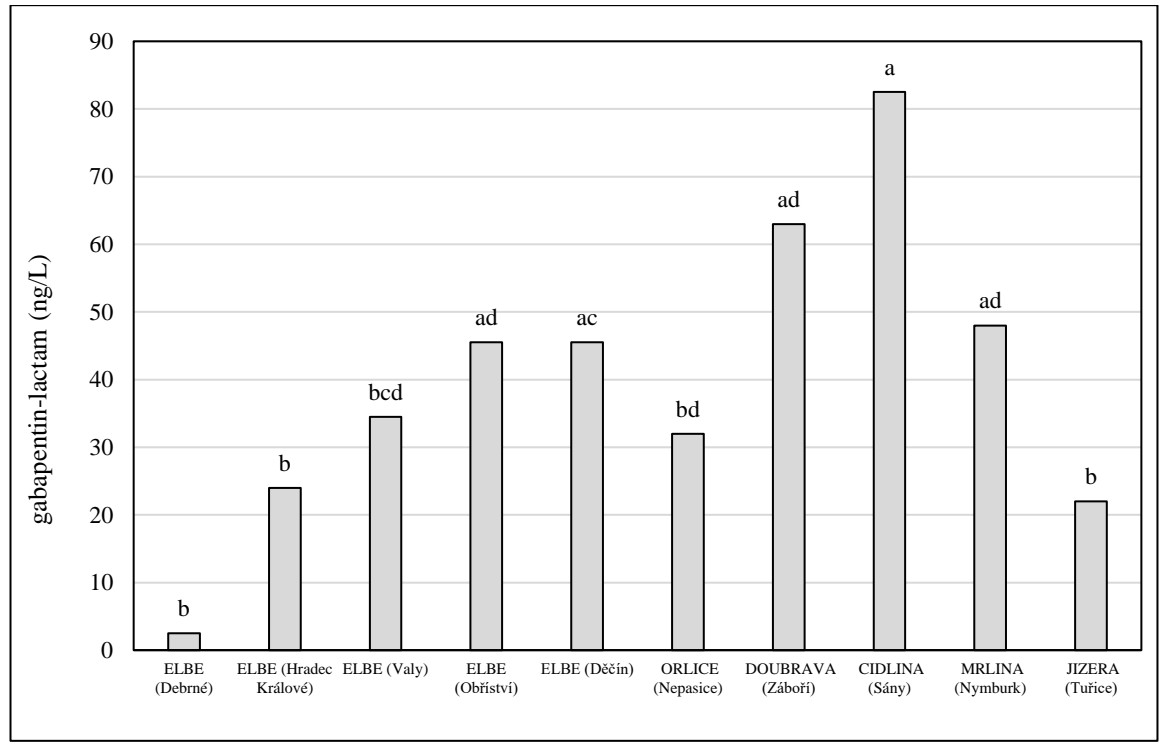

**Figure 9.** Median values of gabapentin-lactam concentration at the sampled locations during the year 2021. Significant differences among the localities are indicated by different alphabetical superscripts ($p < 0.05$).

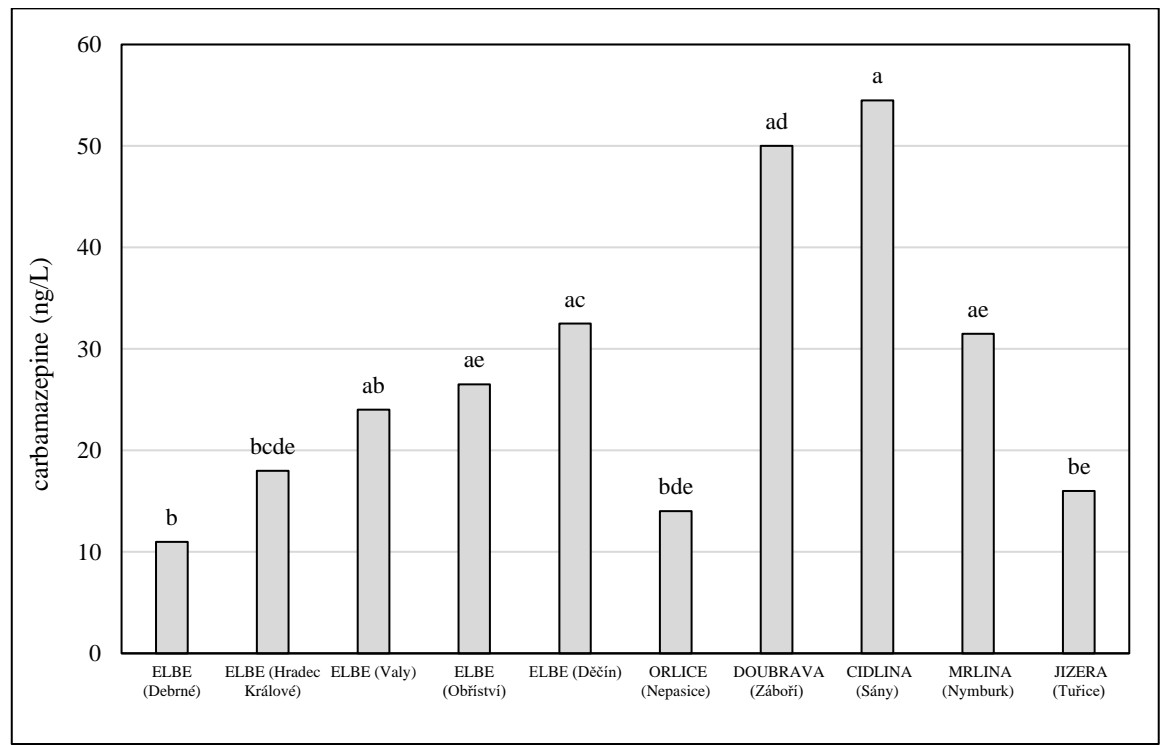

**Figure 10.** Median values of carbamazepine concentration at the sampled locations during the year 2021. Significant differences among the localities are indicated by different alphabetical superscripts ($p < 0.05$).

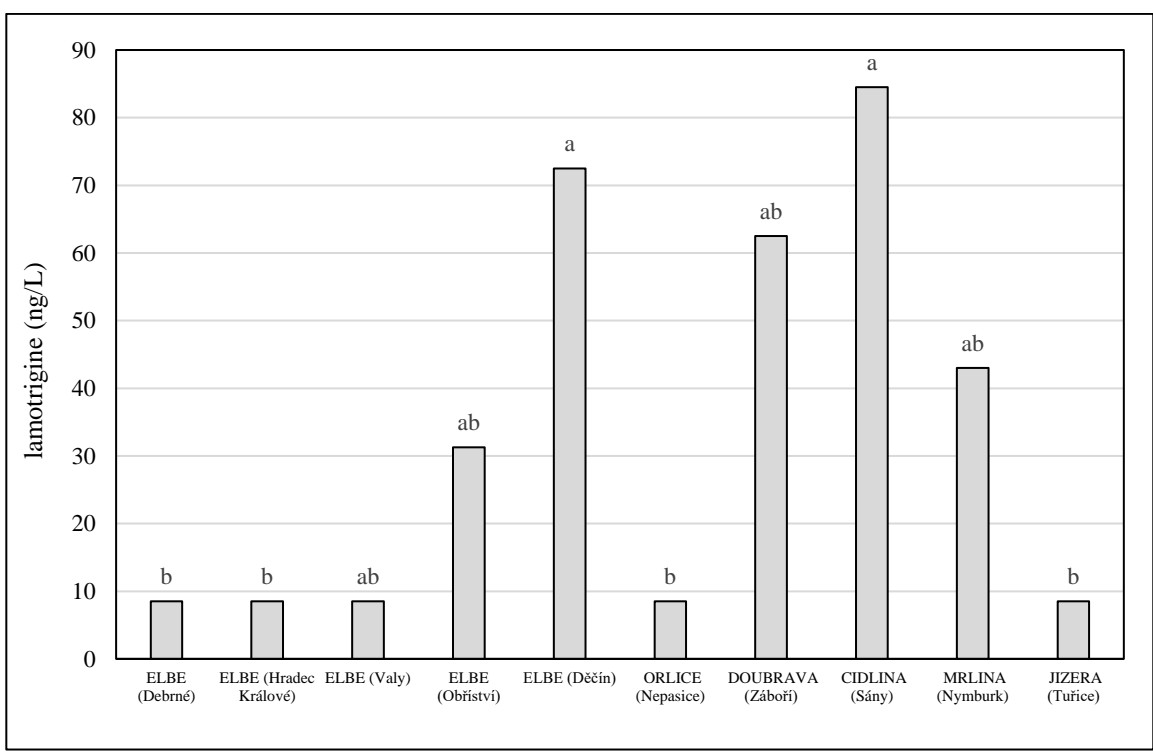

**Figure 11.** Median values of lamotrigine concentration at the sampled locations during the year 2021. Significant differences among the localities are indicated by different alphabetical superscripts ($p < 0.05$).

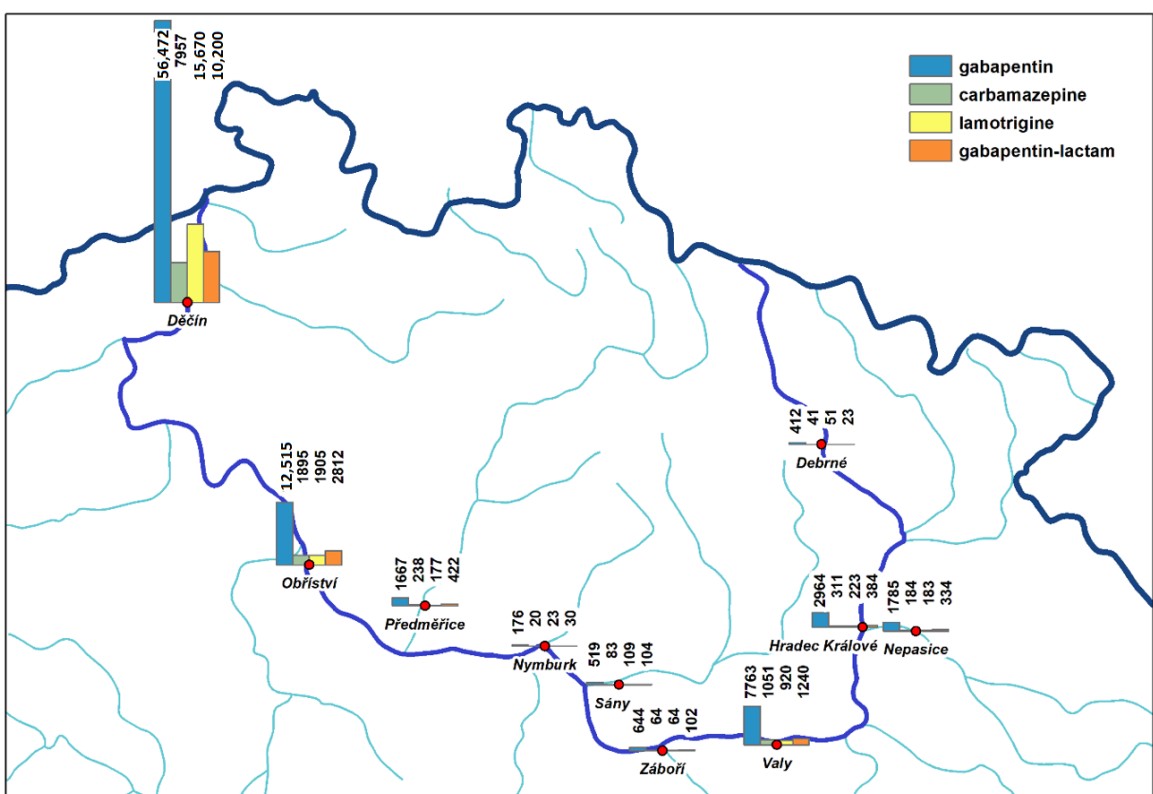

**Figure 12.** Mass flux of anticonvulsants of interest (median value) in µg/s at monitoring sites in 2021.

## 4. Discussion

Residues of pharmaceuticals and their metabolites, which routinely occur at relatively low concentrations (ng/L to low mg/L) in the aquatic environment of developed countries, can still elicit various adverse outcomes on non-target organisms [6] and even contaminate drinking water. In the Czech Republic, the consumption of anticonvulsants is quite high, and the trends and the preference of different substances change over time (as evident from Table 1), based on the increasing knowledge of their efficacy and safety in therapy. Gabapentin is an anticonvulsant used to treat seizures and neuropathic pain that is widely prescribed in many countries [23]. In 2020, 15.9 tons of gabapentin was delivered to pharmacies and medical facilities in the Czech Republic and the amount has been increasing every year (Table 1). Gabapentin is not extensively metabolized in the human body and it is excreted via urine unchanged [24]. Gurke et al. [25] estimated the gabapentin removal rate to be only about 5%. Because of the low elimination of gabapentin in wastewater treatment plants, it is being discharged to rivers and has been detected in surface water samples even at µg/L concentrations [18,26–28]. Ferencik et al. [29] reported gabapentin residues to be up to 210 ng/L in the Vrchlice water reservoir, Czech Republic (maximum volume of 8 millions m$^3$, theoretical retention time of 210 days, supplies water to 60,000 citizens). The water samples were taken close to the collection point of water for the production of drinking water. These results suggest a high persistence of gabapentin. In our study, the lowest median concentration of gabapentin was found in Debrné (79 ng/L). Based on a map of sampling sites (Figure 2), it is evident that this locality is the closest to the Elbe River spring with no big city or other source point in the surroundings. Therefore, it is probably the least contaminated location. The highest concentrations of gabapentin were found in Cidlina (353 ng/L) (Figures 3 and 8). In Děčín (where the Elbe approaches Germany), the median concentration of gabapentin was found to be 257 ng/L.

Surprisingly, Table 2 shows that, in the case of gabapentin, there is no significant correlation between flow rate and concentration, with the exception of the Mrlina locality. Debrné (close to Elbe spring), Jizera, (tributary flowing from ecologically clean localities)

and Orlice were found to be significantly less contaminated by gabapentin residue in comparison with Doubrava, Mrlina and Cidlina (all three are Elbe tributaries will small flow rates and tendencies for substance concentration—Figures 7 and 8). Gabapentin-lactam, a gabapentin analogue, is widely used as a precursor in gabapentin synthesis, even though it possesses some potentially biological activities itself; e.g., it is a $K^+$ channel activator, neuroprotective and neurotrophic [30]. Based on the data in Figure 9, the tributaries Doubrava, Mrlina and Cidlina and last two sampling spots on Elbe (Obříství and Děčín) were again the most contaminated localities (Figures 4 and 9). Since gabapentin is not being produced in the Czech Republic, the source of gabapentin-lactam residues in surface waters of the Elbe River might be the chemical degradation of gabapentin by intramolecular cyclization to gabapentin-lactam and due to impurities in gabapentin-based pharmaceuticals. Carbamazepine is an anticonvulsant drug prescribed worldwide for the treatment of epilepsy, bipolar disorder and trigeminal neuralgia [31]. In the Czech Republic, 3.5 tons of carbamazepine were delivered to pharmacies and medical facilities in 2020 (Table 1). Carbamazepine is almost completely metabolized in the liver with only around 5% of the drug excreted unchanged [32] suggesting that there are other significant routes of carbamazepine to surface waters than just excretion from the body, such as improper disposal of out-of-date medicine to the sewer. The investigations reviewed by Zhang et al. [33] found that carbamazepine is highly persistent, having a removal efficiency usually below 10% by wastewater treatment plants. Zenker et al. [34] and Caracciolo et al. [35] reported carbamazepine to have low elimination rates by wastewater treatment plant processes (<50%) and slow degradation in aquatic environments (half-life: $82 \pm 11$ days). As a consequence, carbamazepine has been detected in wastewater treatment plant effluents, surface waters, groundwater and occasionally in drinking water all over the world [35]. Golovko et al. [10] reported a negative removal efficiency ($-12\%$) after monitoring carbamazepine in a wastewater treatment plant in České Budějovice, Czech Republic, where the median concentration of carbamazepine was 0.46 µg/L in influent water and 0.51 µg/L in effluent water. In our study, the lowest concentrations of carbamazepine were assessed in the locality Debrné (median value 12 ng/L), and, again, the highest in the locality Cidlina with a median value of 54.5 ng/L. Figures 5 and 10 show that in the case of the Elbe River itself (tributaries excluded), the further the sampling site is from the spring the higher the carbamazepine contamination is (Debrné < Hradec Králové < Valy< Obříství < Děčín). In Děčín, where the Elbe approaches Germany, the median concentration was estimated to be 32.5 ng/L, which means that slowly degradable substances tend to accumulate and rise along the river stream (concentrations closer to the spring are lower than the concentrations down the stream). Our results are in accordance with the results of ter Laak et al. (2010) [36], who monitored a large dataset of pharmaceuticals at various sampling locations along the river Rhine over a period of 7 years. The special trends of this study show that the concentration of carbamazepine tends to increase from the river spring towards the Rhine delta.

Regarding the tributaries of the river Elbe, samples taken in Cidlina (location with the highest measured concentration) contained significantly higher concentrations of carbamazepine in comparison with Debrné, Orlice and Jizera (Figures 5 and 10). Based on the data given in Table 2, there is a significant correlation between flow rate and concentration for carbamazepine in all sampling sites. Therefore, it is probably the very small flow rate in Cidlina (Figure 7), as well as the direct contamination from sewage waters (due to absence of wastewater treatment plants on this tributary) that are responsible for the higher carbamazepine concentrations in several big cities and some settlements. In contrast, the Jizera and Orlice springs are in an ecologically clean landscape, so these tributaries might be diluted by the water from clean upper streams and therefore contain lower concentrations of pharmaceutical residues even though they are flowing through cities further downstream. Skocovska et al. [37] analyzed the presence of residues of selected sulfonamides, non-steroidal anti-inflammatory drugs and analgesics-antipyretics in the surface water of the Elbe River basin. In their study, the correlation analysis showed a statistically significant correlation between the river flow rates and the concentration of the drug residues of

ibuprofen, naproxen, diclofenac and paracetamol. In the case of ibuprofen, naproxen and diclofenac, the sampling point located in the river Cidlina was among the most contaminated locations selected for the study. Lamotrigine is an anticonvulsant used on its own or in combination with carbamazepine for the treatment of epilepsy. In 2020, the delivery of lamotrigine to pharmacies and medical facilities in the Czech Republic amounted to 1866 kg and the trend of usage is increasing every year (Table 1). Approximately 10% of lamotrigine is excreted from the human body intact and enters the aquatic environment via wastewater treatment plants [38]. Writer et al. [39] reported lamotrigine concentration in wastewater effluent and receiving waters to be $520 \pm 320$ ng/L. Similar to carbamazepine, lamotrigine was also shown to have a negative removal efficiency in wastewater treatment plants (about $-170\%$) [40]. In our study, the median concentration of lamotrigine in Debrné was found to be 22 ng/L, and Cidlina was again the most contaminated location with a median concentration of 84.5 ng/L. Median concentration in Děčín was 73 ng/L (Figures 6 and 11). In case of lamotrigine, we found a correlation between flow rate and concentration for almost all sampling spots with the exception of Jizera (Table 2). Děčín and Cidlina were found to be the most contaminated localities (Figure 11). The data on mass flux (Figure 12) support the above-described data presented in Figures 3–11. In Děčín, where the mass flux is dominant among the other sampling locations, the gabapentin value is of special interest since the mass flux of this substance is the largest (with 132,335 µg/s and is equivalent to 11.44 kg/den, which could reach 2000 kg/year, 13% of the annual consumption). As expected, mass flux of the pharmaceuticals of interest has been the lowest in localities Mrlina, Cidlina and Doubrava, making these small tributaries endangered with high contamination risk and related ecotoxicity for aquatic organisms. Various scientific studies have proven the exposure of gabapentin at environmental concentrations to be the cause of neurotoxicity and oxidative damage to fish [41,42]. Similar to gabapentin, gabapentin-lactam seems to be able to cause adverse effects during the development of the nervous system in fish [43]. In the case of carbamazepine, oxidative stress [44], decreased embryo production, irregularities in oocytes [45] and decreased sperm motility and velocity [46] were reported. To the best our knowledge, no effects of prolonged or chronic exposure to lamotrigine at environmentally relevant concentrations to fish has been published yet.

## 5. Conclusions

The Elbe River basin is the fourth largest in Europe. The river Elbe is a major water resource in the Czech Republic and an important ecological habitat. The Elbe lowland is also a densely populated area with agricultural and industrial significance. As it continues to flow outside the Czech borders, its contamination is of international importance. In the Czech Republic, the consumption of anticonvulsants is quite high due to developed and available healthcare, and together with low removal efficiency of the monitored substances in wastewater treatment plants, it raises concerns about the possible health effects in non-target organisms living in this ecosystem, about the possible contamination of soil and crops irrigated by the water with such contaminants and about the need for their long-term monitoring. Therefore, the aim of the present study was to determine the presence and levels of anticonvulsant pharmaceuticals (carbamazepine, lamotrigine and gabapentin) and the analogue gabapentin (i.e., gabapentin-lactam) in water samples obtained throughout the year 2021 from 10 sampling sites in the Elbe River basin and its tributaries, located in the Czech Republic. Based on the results of this study, it is possible to conclude that the further the sampling point is from the Elbe River spring, the higher the concentrations are of monitored pharmaceuticals. Also, several small tributaries are significantly contaminated due to their low flow rates and local sources of contamination, with the exception of streams flowing from preserved natural sites. These tributaries contribute to the increasing contamination of the Elbe River along its flow. Thus, tributaries with slow flow rates, and therefore lower mass flux, represent higher ecotoxicological risks for aquatic organisms living in them.

**Supplementary Materials:** The following supporting information can be downloaded at: https: //www.mdpi.com/article/10.3390/w14244122/s1, Figure S1: Chromatograms from the analysis; Table S1: List of most important wastewater treatment plants located upstream from indicated sampling points.

**Author Contributions:** Investigation, data curation, reviewing: M.F.; data curation, software: J.B.; analysis, data curation: J.S.; writing, reviewing: Z.S. (Zuzana Siroka); supervision: Z.S. (Zdenka Svobodova); methodology, reviewing: V.K.; data curation: K.S.; writing, reviewing, editing: P.L. All authors have read and agreed to the published version of the manuscript.

**Funding:** This study was supported by the Internal Creative Agency of the University of Veterinary Sciences Brno (Project No. FVHE/Vecerek/ITA2020).

**Institutional Review Board Statement:** Not applicable.

**Informed Consent Statement:** Not applicable.

**Data Availability Statement:** Not applicable.

**Conflicts of Interest:** The authors declare no conflict of interest.

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
