# Peer review of "Residues of Selected Anticonvulsive Drugs in Surface Waters of the Elbe River Basin (Czech Republic)"

_water, doi:10.3390/w14244122_

Round 1
Reviewer 1 Report (New Reviewer)
The manuscript is interesting, monitoring pharmaceuticals in surface water could emerge the concern of decision makers of the policy of water and pharmaceuticals management. The manuscript is well design and written, however some minor comments need to be addressed before its publication.
Minor comments:
-Minor spell check is required
-some typo errors are presents, Lines 201, 421, 448, 474.
Line 202, It would be interessted if you compare these data with others from different countries.
- Line 204, Please move the table's title up.
- Line 213, How did you calculate its concentration?
Author Response
comments to reviewer are given in the attached file

Reviewer 2 Report (New Reviewer)
no comments are need
Author Response
reviewer had no comments
This manuscript is a resubmission of an earlier submission. The following is a list of the peer review reports and author responses from that submission.
Round 1
Reviewer 1 Report
Comments for Residues of selected anticonvulsive drugs in surface waters of the Elbe River basin, by Ferencik et al.
The paper presents the results of a survey, analyzing the concentration of four anticonvulsant pharmaceuticals and their analogue (carbamazepine, lamotrigine, gabapentin and gabapentin-lactam) in the Elbe River and its tributaries located in the Czech Republic.
Although the paper shows (for the first time?) the presence of 4 compounds in the specific river, it does not present any scientific novelty of global interest, nor contribute to the general knowledge. Its outcome is of a local nature with limited scientific interest and should be published as a technical report in a local journal.
Over the past 20 years there has been an exponential growth in the number of papers related to pharmaceuticals in the environment, including detection of anticonvulsants in rivers. Hence, at this point in time, a scientific publication in this field should present a unique view, a novel perspective or a new detection/treatment technique to be published.
Some specific comments:
L91: replace “point water samples” by grab samples
L93: Why were the sampling sites located at these points? A real explanation should be provided (near cities, near wastewater discharge, etc)
L98-100: Sentence not clear.
Figure 1: Please add direction of flow
Section 2.3. Sample and calibration standards preparation: The entire section is overloaded with unnecessary details (e.g. “1 mL of each filtered sample was dispensed by a 1 mL gas-tight Hamilton syringe in two 2 ml TruView™ vials”). The section should be rewritten in a more concise way.
In addition, there is no concentration procedure (SPE, LLE)? I find it extremely hard to believe that samples were directly injected into the HPLC/MS, and no concentration was needed to achieve detection limits in the nanograms per liter.
Table 1 and related discussion: I am not sure what’s the contribution of this table to the study, since the drugs were not monitored in the river over 10 years and compering between the drugs was not in the papers’ goals. Also, between what data does the statistical analysis in lines 186-188 try to correlate?
L190-193 and throughout the text: English should be revised.
L199-209: Should be placed next to the relevant figures. Also, there is no discussion on the difference in drugs concentrations between months.
Figure 7: How was the flow measured?
Table 2: Data relates to a specific month? An average between months?
Figure 8: The letter C is lacking.
Figure 12: A cake-figure is inadequate in the case, since drugs concentrations do not complete each other to a whole.
L480-500: Belong to the Introduction
L508: Cities are marked on the map, so they cannot be mentioned in the discussion
L512-524: No explanation is provided for the abnormality of gabapentin
L545: Not clear
L551-553: What cities, how many, where exactly?
L582-583: The sentence contradict itself
Reviewer 2 Report
Please find it attached here.

Reviewer 3 Report
In this manuscript, the authors aimed to determine the presence and concentrations of some anticonvulsant pharmaceuticals (carbamazepine, lamotrigine, gabapentin and gabapentin-lactam) in water samples obtained from 10 sampling places in the Elbe River and its tributaries located in the Czech Republic, and to evaluate their correlations with the flow rates at the sampling sites in 2021.
The manuscript's subject is original and falls within the journal's scope. However, while there are dozens of first- and second-generations of anticonvulsant drugs in use, such as Clonazepam, Clorazepate, Diazepam, Mephobarbital, Phenobarbital, Phenytoin, Valproic acid and Phenytoin, the fact that only 3 or 4 drugs were included in the study is the weakest point of the study and significantly reduces its scientific value.
1. Lines 64-65: “… anticonvulsant pharmaceuticals (carbamazepine, lamotrigine and gabapentin) and their analogues (gabapentin-lactam) …”. This statement is not true! gabapentin-lactam is only analogue of gabapentin. Should be revised throughout the manuscript.
2. Lines 91-100: It should be stated from which level of the river the water samples were taken. Surface and bottom water samples may differ. In addition, the inclusion of sediment samples in the research would further increase the scientific value of the study.
3. Lines 112-133: It should be stated how and with which solvent the analytical standards were prepared.
How the LOD and LOQ values were determined should be briefly explained.
It should be stated at what concentration and how much volume the isotopic analogue internal standards were added to the samples.
4. Line 129: “… the final concentration of a spiked sample was 100 ng/l)”. What does it mean “spiked sample”? More information should be given for the method development and validation process of the analysis.
5. Line 153: “… was 24.5 min. Injection volume was 250 μL.”. Wasn't this injection volume too much for a UHPLC column and MS analysis?
6. Results: Sample chromatograms for the liquid chromatography-mass spectrophotometry (UPLC-MS) analysis of the analytical standard with IS’s and water samples contaminated with the drugs studied should be added to the manuscript. Each compound should be numbered and specified in the chromatograms.
7. Figures 8, 9, 10 and 11: (+/-) standard deviation values should be added to each column in each table.